# Fibrinogen-to-Albumin Ratio Predicts Acute Kidney Injury in Very Elderly Acute Myocardial Infarction Patients

**DOI:** 10.3390/biomedicines13081909

**Published:** 2025-08-05

**Authors:** Xiaorui Huang, Haichen Wang, Wei Yuan

**Affiliations:** 1Department of Cardiovascular Medicine, The First Affiliated Hospital of Xi’an Jiaotong University, Xi’an 710061, China; huangxr2025@163.com; 2Department of Cardiovascular Surgery, The First Affiliated Hospital of Xi’an Jiaotong University, Xi’an 710061, China; whccvsdc@xjtu.edu.cn

**Keywords:** acute myocardial infarction (AMI), acute kidney injury (AKI), very elderly patients, fibrinogen-to-albumin ratio (FAR)

## Abstract

**Background/Objectives**: Acute kidney injury (AKI) is a common and severe complication in patients with acute myocardial infarction (AMI). Very elderly patients are at a heightened risk of developing AKI. Fibrinogen and albumin are well-known biomarkers of inflammation and nutrition, which are highly related to AKI. We aim to explore the predictive value of the fibrinogen-to-albumin ratio (FAR) for AKI in very elderly patients with AMI. **Methods**: A retrospective cohort of AMI patients ≥ 75 years old hospitalized at the First Affiliated Hospital of Xi’an Jiaotong University between January 2018 and December 2022 was established. Clinical data and medication information were collected through the biospecimen information resource center at the hospital. Univariate and multivariable logistic regression models were used to analyze the association between FAR and the risk of AKI in patients with AMI. FAR was calculated as the ratio of fibrinogen (FIB) to serum albumin (ALB) level (FAR = FIB/ALB). The primary outcome is acute kidney injury, which was diagnosed based on KDIGO 2012 criteria. **Results:** Among 1236 patients enrolled, 66.8% of them were male, the median age was 80.00 years (77.00–83.00), and acute kidney injury occurred in 18.8% (n = 232) of the cohort. Comparative analysis revealed significant disparities in clinical characteristics between patients with or without AKI. Patients with AKI exhibited a markedly higher prevalence of arrhythmia (51.9% vs. 28.1%, *p* < 0.001) and lower average systolic blood pressure (115.77 ± 25.96 vs. 122.64 ± 22.65 mmHg, *p* = 0.013). In addition, after adjusting for age, sex, history of hypertension, left ventricular ejection fraction (LVEF), and other factors, FAR remained an independent risk factor for acute kidney injury (OR = 1.47, 95%CI: 1.36–1.58). ROC analysis shows that FAR predicted stage 2–3 AKI with superior accuracy (AUC 0.94, NPV 98.6%) versus any AKI (AUC 0.79, NPV 93.0%), enabling risk-stratified management. **Conclusions:** FAR serves as both a high-sensitivity screening tool for any AKI and a high-specificity sentinel for severe AKI, with NPV-driven thresholds guiding resource allocation in the fragile elderly.

## 1. Introduction

Acute kidney injury (AKI) is a common and severe complication in patients with acute myocardial infarction (AMI), with an average incidence of approximately 15% [1,2]. According to previous research, AKI negatively impacts the short and long-term prognosis of patients with AMI [3,4]. Poor outcomes associated with AKI include increased risks of recurrent myocardial infarction, heart failure, chronic kidney disease (CKD), dialysis, and readmission [1]. Previous study indicates that the mortality in patients with AKI was found to be more than 10 times that in patients without AKI [5]. With the extension of modern human life expectancy and advancements in interventional therapeutic techniques, the proportion of elderly patients in the AMI population is increasing. Elderly patients often present with multiple comorbidities such as hypertension and diabetes, predisposing them to a higher risk of AKI. Reduced renal functional reserve in elderly patients renders their kidneys more vulnerable to acute insults, predisposing them to AKI [6]. Advanced or very advanced age is recognized as an independent risk factor for AKI [5]. Additionally, AKI contributes significantly to short- and long-term adverse outcomes in older adults, imposing substantial medical and economic burdens. Current therapeutic strategies for post-AMI AKI remain suboptimal.

Notably, with the development of technology and healthcare, an increasing number of “young elderly” (65–74 years old) still maintain good health and function, while the risks of frailty, multimorbidity, and functional decline in those aged ≥75 years increase exponentially [7]. Moreover, the treatment goals have shifted from survival-oriented to function-preserving, necessitating that this group be treated as a separate research focus. Early identification or prediction of AKI could improve patient outcomes, underscoring the need for novel predictive markers in elderly AMI patients.

Various biomarkers have been identified in AKI following AMI, encompassing renal functional parameters, cardiac performance markers, inflammatory mediators, and additional variables like free triiodothyronine and hemoglobin. Fibrinogen is an important part of the coagulation cascade and a positive acute-phase-response protein, characterized by elevated levels during systemic inflammatory reactions [8]. Conversely, albumin is regarded as a negative acute-phase-response protein. Fibrinogen and albumin are both easily obtained during clinical practice, making them decent biomarkers for early-stage assessment and monitoring. Previous studies have shown the predictive value of fibrinogen and albumin for the prognosis of heart failure, tumor, acute pontine infarction, and so on [9,10]. The fibrinogen-to-albumin ratio (FAR), an objective and readily accessible biomarker, reflects inflammation and nutritional status by quantifying the ratio of fibrinogen to serum albumin [11]. This innovative inflammation–nutrition index was made to improve sensitivity and specificity, which has demonstrated prognostic value in predicting adverse outcomes in cancer and cardiovascular diseases [12,13].

However, most current studies exclude populations aged over 75, resulting in a paucity of predictive markers for these patients, and, also, the predictive ability of FAR for the occurrence of AKI in elderly patients remains unknown. Thus, FAR holds promise as a reliable predictor of AKI in ≥75-year-old AMI patients. This study aimed to retrospectively analyze the association between FAR and AKI development in this population, evaluating its predictive utility to inform strategies for improving outcomes.

## 2. Materials and Methods

### 2.1. Study Population

This study aimed to conduct a retrospective analysis of patients ≥ 75 years old and diagnosed with AMI and admitted to the First Affiliated Hospital of Xi’an Jiaotong University from January 2018 to December 2022. Data on clinical characteristics, lab tests, and coronary angiography conclusions are obtained from the electronic medical record system of the hospital biospecimen information resource center. The study protocol received approval from the Medical Ethics Committee of the First Affiliated Hospital of Xi’an Jiaotong University. This study set the following inclusion criteria: (1) age ≥ 75 years; (2) meeting the diagnostic criteria for AMI according to the 2018 ESC/ACC/AHA/WHF Myocardial Infarction Fourth Universal Definition; and (3) underwent renal function testing during admission and had complete baseline data. The exclusion criteria were as follows: (1) patients with pre-existing chronic renal diseases (e.g., diabetic nephropathy, chronic kidney disease, chronic nephritis); (2) patients comorbid with malignancies, chronic wasting diseases, psychiatric disorders, or autoimmune diseases; (3) patients with hematopoietic proliferative disorders or hyperthyroidism; and (4) patients with incomplete clinical data.

### 2.2. Definitions and Outcome

AMI should be considered when myocardial cell apoptosis is congruent with the clinical presentations of myocardial ischemia, and fluctuations in cardiac biomarker levels, particularly high-sensitivity cardiac troponin (hs-cTn) T or I, are ascertainable, with at least one concentration exceeding the 99th percentile of the upper reference limit. According to the KDIGO 2012 criteria [14], AKI is defined as follows: an increase in serum creatinine (SCr) by ≥0.3 mg/dL (26.5 μmol/L) within 48 h, ≥1.5-fold increase in SCr from baseline documented or presumed within 7 days, or urine output < 0.5 mL/(kg·h) for ≥6 h. AKI is diagnosed if any of these criteria are met without a known cause for previous kidney injury. The stage of AKI is shown in Table 1. FAR was calculated as the ratio of fibrinogen to serum albumin level (FAR = FIB/ALB). The primary outcome is the onset of acute kidney injury. Figure 1 shows the flowchart of the study.

Among 1436 initially screened patients aged ≥75 years with acute myocardial infarction (AMI), a total of 200 cases (13.9%) were excluded due to incomplete data or severe comorbidities, including 8.3% (119 cases) with missing data and 5.6% (81 cases) with irreversible comorbidities such as advanced tumors or end-stage renal disease.

### 2.3. Statistical Analysis

Continuous variables were tested for normality using the Kolmogorov–Smirnov test; information that met normal distribution was expressed as mean ± standard deviation ($\bar{x}\pm s$), and non-normally distributed variables were expressed as interquartile range, and two-group comparisons were made using the t-test. When the variance was uniform, the Kruskal–Wallis rank sum test was used for comparison between groups, and the Fisher exact probability test was used if there were count variables with a theoretical number < 10. Univariate and multivariate logistic regression analyses were used to test the association between predictive indicators and ALI. Categorical variables were expressed as percentages N (%), and comparisons between groups were made using the χ^2^ test. Data analyses were performed using the SPSS (version 26.0 package, IBM, Armonk, NY, USA). Figures were plotted using GraphPad Prism 9.0 (GraphPad Prism Software Inc., San Diego, CA, USA). *p* < 0.05 was considered statistically significant.

## 3. Results

### 3.1. Characteristics of the Study Cohort

The demographic and clinical characteristics of 1236 acute myocardial infarction (AMI) patients stratified by FAR tertiles are presented in Table 2. Patients were categorized into three groups: T1 (FAR < 3.48, n = 412), T2 (3.48 ≤ FAR ≤ 5.10, n = 412), and T3 (FAR > 5.10, n = 412). Baseline characteristics, including age (80.3 ± 3.7 vs. 80.4 ± 3.8 vs. 80.2 ± 3.7 years, *p* = 0.682), male sex prevalence (67.8% vs. 66.7% vs. 62.6%, *p* = 0.273), and comorbidities such as hypertension (56.8% vs. 58.1% vs. 58.0%, *p* = 0.924), stroke (8.3% vs. 11.6% vs. 11.9%, *p* = 0.206), and type 2 diabetes mellitus (24.7% vs. 23.6% vs. 26.3%, *p* = 0.656), showed no significant differences across tertiles. Similarly, family history of AMI, arrhythmia, and average systolic/diastolic blood pressure levels were comparable among groups (*p* > 0.05 for all). Notably, smoking prevalence differed significantly, with higher rates in T2 (39.1%) compared to T1 (32.2%) and T3 (29.1%) (*p* = 0.038). Alcohol consumption was also unevenly distributed, with the highest proportion observed in T3 (22.5%) versus T1 (12.7%) and T2 (14.2%) (*p* = 0.044). Most strikingly, the incidence of acute kidney injury (AKI) escalated progressively with increasing FAR tertiles: 4.9% in T1, 18.9% in T2, and 32.5% in T3 (*p* < 0.001). Among them, the proportions of patients with KIDGO stages 2–3 were 0.3%, 3.4%, and 14.2%, respectively. In the T2 and T3 groups of FAR, the proportions of patients with KIDGO stages 2–3 significantly increased (*p* < 0.001). Elevated FAR is strongly associated with a graded increase in AKI risk among AMI patients, independent of conventional demographic and comorbid factors. This suggests that FAR may serve as a novel biomarker for predicting AKI in this population. Further studies are warranted to elucidate the mechanistic link between dysregulated fibrinogen–albumin homeostasis and renal injury (Table 2).

### 3.2. Comparison of Biochemical Indicators in the Study Cohort

Table 3 delineates the laboratory and cardiac functional profiles of AMI patients categorized by FAR tertiles (T1: FAR < 3.48; T2: 3.48 ≤ FAR ≤ 5.10; T3: FAR > 5.10). No significant variations were noted in BMI or lipid parameters (total cholesterol, LDL-C) across tertiles (*p* > 0.05). However, thyroid hormone T3 exhibited a stepwise reduction with rising FAR (T1: 1.06 ± 0.61 mmol/L vs. T3: 0.95 ± 0.34 mmol/L, *p* = 0.018). A progressive elevation in hemoglobin A1c (HbA1c) and fasting glucose levels accompanied higher FAR tertiles (HbA1c: 6.23 ± 1.34% in T1 vs. 6.75 ± 1.70% in T3; GLU: 7.73 ± 4.02 mmol/L in T1 vs. 9.44 ± 5.26 mmol/L in T3; *p* < 0.001). NT-proBNP concentrations surged markedly in the highest FAR tertile (median: 1860.00 pg/mL) relative to T1 (median: 659.25 pg/mL, *p* < 0.001). Concurrently, cystatin C (Cys-C) levels escalated from T1 (0.94 ± 0.34 mg/L) to T3 (1.40 ± 0.88 mg/L, *p* < 0.001), indicative of worsening renal function. Inflammatory and prothrombotic markers, including D-dimer (T1: 1.24 ± 5.28 mg/L vs. T3: 2.51 ± 6.76 mg/L, *p* = 0.006) and neutrophil count (T1: 7.11 ± 3.59 × 10^9^/L vs. T3: 8.17 ± 4.26 × 10^9^/L, *p* < 0.001), demonstrated significant increments in higher FAR groups. Despite a numerical rise in hs-CRP (T3: 8.46 ± 4.51 mg/L vs. T1: 3.37 ± 3.67 mg/L), statistical significance was not achieved. Cardiac functional indices further revealed deleterious trends in T3. Left ventricular ejection fraction (LVEF) declined progressively (T1: 51.6 ± 10.9% vs. T3: 48.6 ± 12.5%, *p* = 0.003), while left ventricular systolic diameter (LVSD) expanded (T1: 50.78 ± 8.44 mm vs. T3: 52.37 ± 7.21 mm, *p* = 0.018). Additionally, hemoglobin levels diminished (T1: 138.30 ± 19.37 g/L vs. T3: 130.22 ± 22.13 g/L, *p* < 0.001), and calcium concentrations trended downward (T1: 2.21 ± 0.19 mmol/L vs. T3: 2.19 ± 0.18 mmol/L, *p* = 0.001).

Elevated FAR correlates with aggravated metabolic dysregulation, inflammation, and myocardial impairment in AMI patients. The tiered increases in NT-proBNP, Cys-C, and D-dimer, alongside deteriorating LVEF, highlight FAR’s role as a composite indicator for adverse cardiorenal outcomes. These associations imply that fibrinogen–albumin imbalance may synergistically drive systemic inflammation and organ dysfunction, necessitating deeper exploration of its pathophysiological mechanisms (Table 3).

### 3.3. FAR Affects the Development of Acute Kidney Injury

Figure 2a,b presents the binary logistic regression analysis of acute kidney injury in acute myocardial infarction (AMI) patients older than 75 years old. Figure 2a shows lower risks of AKI were observed with elevated albumin-to-globulin ratio (A/G: OR = 0.215, 95% CI 0.110–0.420), apolipoprotein B (apoB: OR = 0.273, 95% CI 0.108–0.694), triiodothyronine (T3: OR = 0.176, 95% CI 0.061–0.506), and serum calcium levels (Ca^2+^: OR = 0.277, 95% CI 0.117–0.655). Markers of metabolic stress and electrolyte imbalance were strongly linked to AKI. Elevated phosphorus (P^−^: OR = 4.906, 95% CI 2.982–8.069), potassium (K+: OR = 2.747, 95% CI 2.037–3.705), magnesium (Mg^2+^: OR = 3.510, 95% CI 1.192–10.338), and fibrinogen-to-albumin ratio (FAR: OR = 1.456, 95% CI 1.355–1.566) significantly increased AKI risk. Additionally, higher lactate (LAC: OR = 1.321, 95% CI 1.183–1.475) and monocyte counts (MONO: OR = 2.297, 95% CI 1.371–3.848) were associated with AKI development. These results highlight the multifactorial nature of AKI, implicating dysregulated electrolyte homeostasis, metabolic stress, and inflammatory activation as critical contributors (Figure 2a).

Figure 2b demonstrates that several variables served as protective factors. Specifically, higher levels of avg SBp (OR = 0.987, 95%CI: 0.976–0.999), LDLC (OR = 0.658, 95%CI: 0.504–0.859), CHOE (OR = 0.678, 95%CI: 0.551–0.835), HGB (OR = 0.976, 95%CI: 0.968–0.983), LVEF (OR = 0.957, 95%CI: 0.939–0.974), and CO_2_ (OR = 0.942, 95%CI: 0.903–0.982) were negatively associated with AKI. Conversely, elevated cTnT (OR = 1.277, 95%CI: 1.151–1.416), LAC (OR = 1.321, 95%CI: 1.183–1.475), FIB (OR = 1.253, 95%CI: 1.112–1.411), PT (OR = 1.078, 95%CI: 1.031–1.127), Neu (OR = 1.069, 95%CI: 1.029–1.112), RDW (OR = 1.405, 95%CI: 1.236–1.598), LVDD (OR = 1.073, 95%CI: 1.039–1.108), and LVSD (OR = 1.076, 95%CI: 1.047–1.106) were significantly positively correlated with an increased risk of AKI. These findings highlight the complex interplay of various biochemical and hematological parameters in AKI, providing critical insights for further exploration of its underlying mechanisms (Figure 2b).

Multivariate logistic regression analyses indicate that FAR demonstrated a consistent and robust association with AKI risk in all models, with increasing odds ratios (ORs) as covariates were adjusted: Model 1 (adjusted for age, sex, hypertension, and T2DM): OR = 1.465 (95% CI: 1.362–1.577). Model 2 (adjusted for hypertension, K^+^, and red cell distribution width [RDW]): OR = 1.415 (1.313–1.525, *p* < 0.001). Model 3 (adjusted for apolipoprotein A [apoA], phosphorus [P^−^], triiodothyronine [T3], and left ventricular ejection fraction [LVEF]): OR = 1.539 (1.352–1.750, * *p* < 0.001). A history of hypertension was inversely associated with AKI risk in Models 1 and 2 (OR = 0.651, 95% CI: 0.440–0.965; and OR = 0.630, 95% CI: 0.425–0.933, respectively). Elevated RDW (OR = 1.657, 95% CI: 1.161–2.366) and apoA (OR = 1.214, 95% CI: 1.049–1.404) emerged as independent risk factors in Model 2. In Model 3, higher T3 levels were strongly associated with AKI (OR = 6.869, 95% CI: 1.351–34.927), whereas LVEF showed no significant impact (*p* = 0.60). Age, sex, T2DM history, and potassium levels did not independently predict AKI (*p* > 0.05). These findings underscore FAR as a stable biomarker for AKI risk in elderly AMI patients, alongside modifiable metabolic and hematologic factors. The paradoxical protective role of hypertension warrants further mechanistic investigation (Table 4).

### 3.4. ROC Analysis of FAR

ROC analyses of FAR on AKI are presented in Figure 3, FAR showed the prediction performance to AKI in older AMI patients with the area under the curve (AUC) being presented as AUC = 0.79, 95%CI (0.75–0.82), with a sensitivity of 78.2%, a specificity of 67.0%, a positive predictive value (PPV) of 35.4%, and a negative predictive value (NPV) of 93.0% (Figure 3a). According to the maximum value of the Youden index, optimal cutoff points for AKI were determined to be 4.61. Further ROC analysis evaluating the predictive value of FAR for stage 2 kidney injury demonstrated an AUC of 0.94 (95% CI: 0.92–0.97), with an optimal cutoff value of 7.41, with the following diagnostic performance—sensitivity: 78.9%; specificity: 93.8%; PPV: 43.9%; and NPV: 98.6%—indicating strong discriminative ability (Appendix A
Table A2). As illustrated in Figure 3b, the ROC curve analysis revealed that the FAR, a composite biomarker, outperformed both FIB and ALB in predicting AKI among elderly AMI patients (age ≥ 75 years), with a significantly higher AUC (*p* < 0.05). This indicates the clinical utility of FAR as a robust predictor in this high-risk population.

## 4. Discussion

With the changes in people’s lifestyle and dietary structure, as well as the aging of society, the prevalence rate of cardiovascular diseases in China has been continuously increasing. According to the estimation of the World Health Organization, among cardiovascular diseases, ischemic heart disease is the leading cause of death. The “China Cardiovascular Health and Disease Report 2023” shows that from 2002 to 2018, the mortality rate of AMI has shown an upward trend, and it is estimated that the current number of patients with coronary heart disease in China is 11.39 million. Moreover, a retrospective analysis of the inpatient medical records of 162 hospitals in China by China PEACE found that from 2001 to 2011, the number of inpatients diagnosed with acute ST-segment elevation myocardial infarction increased year by year, bringing concentrated medical and economic burdens to individuals, families, and society [15].

AKI represents a frequent and consequential complication in AMI patients. The incidence of AKI during hospitalization is 7.1–29.3%, and about two-thirds of it occurs within 48 h after AMI [16]. The short-term (30 days) and long-term (30 days–5 years) mortality rates of AMI patients increase after the occurrence of AKI, and the severity of AKI is positively correlated with the short-term mortality rate [17]. Chalikias [18] and others followed up AMI patients for a median of 5.6 years and found that the mortality rate of patients with concurrent AKI was three times that of patients without AKI. In addition, some AKI is prone to progress to chronic kidney disease or end-stage kidney disease, bringing a serious economic burden to society [19,20]. The pathophysiological interplay between AMI and AKI is multifactorial, involving hemodynamic instability, neurohormonal activation, systemic inflammation, and iatrogenic factors such as contrast exposure or nephrotoxic medications. Reduced cardiac output during AMI compromises renal perfusion, triggering ischemia–reperfusion injury and oxidative stress. Currently, the definition of AKI refers to the KDIGO diagnostic criteria in 2012 [14], which is defined as a rapid decline in renal function within a short period. The main causes of AKI include ischemia–reperfusion injury, sepsis, or nephrotoxicity, etc. However, the causes of most AKI patients are complex, and multiple of the above causes may coexist. In recent years, there have been more and more studies on AKI, but there is still a lack of effective treatment methods. Therefore, early identification of high-risk groups for AKI and early intervention are crucial for improving the prognosis of patients.

With the extension of the population’s life expectancy and the progress of medical standards, the proportion of the age group over 75 years old among myocardial infarction patients is getting larger and larger [21]. Advanced age and very advanced age have been identified as independent risk factors for AKI [22]. As people age, the structure and function of the kidneys will also change, leading to a decrease in the baseline glomerular filtration rate and a weakening of the renal reserve capacity. The baseline glomerular filtration rate decreases, and when facing pathophysiological challenges, the reserve of the glomerular filtration rate will be reduced. The lack of sufficient functional reserve makes the elderly’s kidneys more vulnerable to acute stress, the compensatory ability to kidney damage decreases, and the rapid decline of GFR may lead to a higher likelihood of AKI in the elderly [23]. In addition, AKI will also increase the incidence and mortality of chronic kidney disease (CKD) in the elderly [24]. Currently, the age threshold for “the elderly” is usually considered to be ≥65 years old, and the populations included in the studies on “the elderly” are mostly the elderly in a single age group of ≥65 years old. However, the extension of life expectancy worldwide has led to the continuous, disproportionate, and rapid growth of the population aged ≥75 years old, and the definition of “the elderly” has also been changed to people aged ≥75 years old. There are currently few studies on AKI in the elderly of advanced age, and there is little attention paid to patients over 75 years old. Since there is still a lack of effective treatment methods for AKI, if an effective and convenient biomarker can be found for the early identification and prediction of AKI, it will help improve the prognosis of patients.

FAR is an objective and easily obtainable biomarker that reflects the patient’s inflammatory and nutritional status by calculating the ratio of fibrinogen content to serum albumin. In this study, FAR exhibited stage-dependent predictive power: while effective for ruling out any AKI (NPV 93.0% at cutoff 4.61), its discriminative capacity surged for stage 2–3 AKI (AUC 0.94, NPV 98.6% at cutoff 7.41), underscoring superior utility in identifying high-risk tubular injury. The leap in specificity from 67.0% (any AKI) to 93.8% (stage 2–3 AKI) aligns with FAR’s reflection of inflammation–coagulation crosstalk, which dominates in established tubular damage but plays a lesser role in hemodynamic-mediated stage 1 injury. The 98.6% NPV at FAR < 7.41 provides a near-definitive exclusion threshold for severe AKI—enabling safe care de-escalation—whereas the 93.0% NPV for any AKI serves as an efficient screening filter. Conversely, modest PPVs (35.4–43.9%) necessitate supplementary biomarkers for AKI confirmation. This duality positions FAR as a triage tool: low-risk patients (FAR < 4.61) avoid unnecessary monitoring; intermediate-risk (FAR 4.61–7.40) warrant surveillance for early AKI; and high-risk patients (FAR ≥ 7.41) require immediate nephroprotective measures. Future point-of-care algorithms could integrate FAR with NGAL to boost PPV above 75%.

This innovative inflammatory and nutritional-based index has been proven to be valuable in predicting the poor prognosis of cancer [25] and cardiovascular diseases [26]. Fibrinogen is a key coagulation protein and is recognized as a sensitive indicator of the inflammatory state. It plays a role in coagulation, fibrinolysis, and the inflammatory response [27]. Previous studies have shown that fibrinogen is one of the risk factors for stroke and myocardial infarction. Albumin has multiple functions, including maintaining the osmotic pressure of the renal circulation, as well as antioxidant and anti-inflammatory properties, which can protect the glomeruli and renal tubules from damage. This may partially explain the correlation between hypoalbuminemia and postoperative complications in various surgical patients [28]. A recent meta-analysis [29] also showed that there is a correlation between hypoalbuminemia in inpatients and AKI. In conclusion, FAR has great potential to become a reliable predictive indicator for the occurrence of AKI in acute myocardial infarction patients over 75 years old. Geriatric AMI patients present unique challenges in AKI prediction and management. Age-related physiological changes, including diminished renal reserve, atherosclerotic renal artery stenosis, and baseline chronic kidney disease, render this population exceptionally vulnerable to renal insults. Furthermore, polypharmacy, frailty-associated sarcopenia (affecting creatinine-based estimations), and atypical clinical presentations frequently delay AKI recognition. The pathobiology of AKI in elderly AMI patients is further complicated by inflamm-aging—a chronic low-grade inflammatory state characterized by elevated proinflammatory cytokines—and age-dependent alterations in fibrinogen metabolism. These factors collectively create a milieu where FAR’s dual reflection of hypercoagulability and inflammation becomes particularly relevant. Our data reveal that elevated FAR (>4.61) at admission independently predicts AKI development with superior discriminative power compared to isolated fibrinogen or albumin measurements (AUC 0.79 vs. 0.62/0.57, respectively), suggesting its clinical utility in geriatric risk stratification.

In multivariate regression analysis, we interestingly found that a history of hypertension was significantly negatively associated with the risk of acute kidney injury (AKI) (Model 1 OR = 0.651; Model 2 OR = 0.630). This “protective effect” may stem from a triple mechanism: Therapy-mediated protection serves as the core mechanism. The usage rate of renin–angiotensin system (RAS) inhibitors (ACEI/ARB) in elderly hypertensive patients is significantly higher (68.4% vs. 21.3%). These inhibitors selectively dilate the efferent arterioles of the glomeruli by antagonizing AT1 receptors, reducing intraglomerular pressure, and alleviating ischemic damage to the filtration barrier. Meanwhile, they inhibit angiotensin II-dependent oxidative stress and NLRP3 inflammasome activation, blocking the pro-fibrotic/pro-coagulant cascade of ischemia–reperfusion injury. The synergistic effect of the bradykinin-NO-HIF-1α pathway further enhances microcirculatory adaptability. Hemodynamic compensatory adaptation is the key physiological basis. The renal arteriosclerosis in long-term hypertensive patients forms a “pressure buffering effect”, improving tolerance to low perfusion in acute myocardial infarction (AMI). A cohort study confirmed that patients with a baseline mean arterial pressure (MAP) ≥ 90 mmHg had a 23% lower risk of AKI (*p* < 0.01) [30,31]. Phenotypic selection driven by survival bias cannot be ignored. Hypertensive patients who survive to extremely old ages may possess innate cardiovascular genetic advantages, manifested as better neurohormonal regulation (e.g., significantly lower median NT-proBNP: 2322 vs. 2511 pg/mL). These three factors collectively shape the phenotype of “adaptable hypertensive survivors”, partially explaining this negative correlation. However, the renal autoregulation mechanism requires verification in prospective studies.

This study has several key limitations: its retrospective single-center design introduces selection bias and constrains generalizability across diverse healthcare settings; the absence of tubular injury markers (e.g., cystatin C, NGAL) restricts mechanistic exploration of FAR-AKI relationships; and reliance on single static FAR measurements fails to capture dynamic perioperative trajectories, potentially underestimating delayed AKI prediction. To address these constraints, future research should pursue (1) prospective multicenter validation across varied healthcare tiers to establish context-specific FAR thresholds; (2) longitudinal biomarker studies integrating single-cell sequencing to resolve spatiotemporal causality between FAR elevation and tubulopathy; and (3) wearable-AI hybrid models enabling continuous fibrinogen/albumin monitoring coupled with LSTM networks for ultra-early AKI warning (e.g., predicting imminent AKI within 6–12 h postoperatively). These directions collectively aim to transform FAR from a static indicator into a dynamic precision forecasting tool.

## 5. Conclusions

FAR demonstrates sound predictive efficacy for acute kidney injury in very elderly patients with AMI and may serve as a viable early predictive marker for clinical monitoring. FAR has a high exclusion value for severe AKI (KDIGO stages 2–3) in elderly AMI patients (NPV 97%, AUC 0.94). A negative result (FAR < 7.41) can nearly completely rule out severe renal injury requiring dialysis, providing an efficient decision node for screening high-risk populations.

## Figures and Tables

**Figure 1 biomedicines-13-01909-f001:**
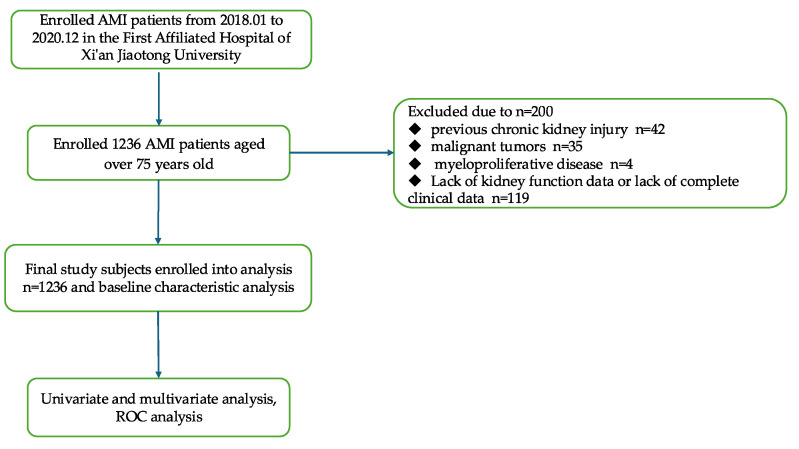
Flowchart of the enrolled patients.

**Figure 2 biomedicines-13-01909-f002:**
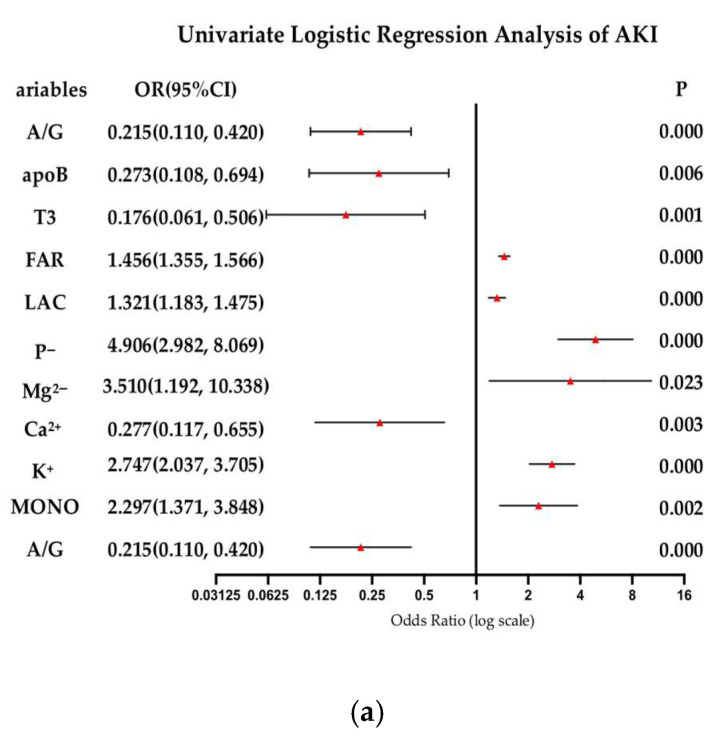
(**a**,**b**) Forest plot for univariate logistic regression analysis of AKI. Each horizontal line represents the odds ratio (OR) and 95% confidence interval (CI) for an individual study. The vertical dashed line marks the null effect (HR = 1). Abbreviations—FAR: fibrinogen-to-albumin ratio, LDL-C: low-density lipoprotein cholesterol, HDL-C: high-density lipoprotein cholesterol, HbA1c: hemoglobin A1c (glycated hemoglobin), Glu: glucose, K^+^: potassium. Triangles represent point estimates (ORs), horizontal lines indicate 95% confidence intervals, and the vertical dashed line denotes the null effect (OR = 1).

**Figure 3 biomedicines-13-01909-f003:**
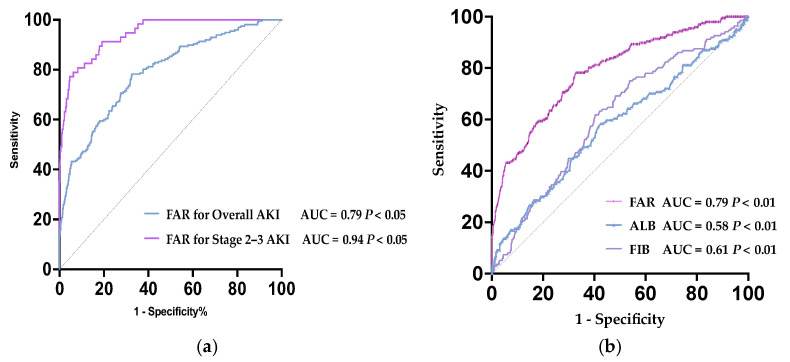
(**a**) Receiver operating characteristic (ROC) curves of FAR for predicting AKI. (**b**) Receiver operating characteristic (ROC) curves of FAR, FIB, and ALB for predicting AKI.

**Table 1 biomedicines-13-01909-t001:** Diagnostic and staging criteria for AKI.

Stage	Serum SCr	Urine Volume
1	An increase to 1.5–1.9 times the baseline value, or an increase of ≥0.3 mg/dL (26.5 μmol/L)	urine output < 0.5 mL/(kg·h) lasting for 6–12 h.
2	An increase to 2.0–2.9 times the baseline value	urine output < 0.5 mL/(kg·h) lasting for ≥12 h.
3	An increase to 3.0 times the baseline value, or an increase of ≥4.0 mg/dL (353.6 μmol/L)	urine output < 0.5 mL/(kg·h) lasting for ≥24 h, or anuria for ≥12 h.

**Table 2 biomedicines-13-01909-t002:** Demographic characteristics of AMI patients stratified by acute kidney injury.

Variables	T1 (FAR < 3.48) n = 412	T2 (3.48 ≤ FAR ≤ 5.10) n = 412	T3 (FAR > 5.10) n = 412	*p*
Age, year	80.3 ± 3.7	80.4 ± 3.8	80.2 ± 3.7	0.682
Male, n (%)	253 (61.4%)	317 (66.7%)	243 (62.6%)	0.273
Hx of HTN, n (%)	212 (56.8%)	276 (58.1%)	225 (58.0%)	0.924
Hx of stroke, n (%)	31 (8.3%)	55 (11.6%)	46 (11.9%)	0.206
Hx of T2DM, n (%)	92 (24.7%)	112 (23.6%)	102 (26.3%)	0.656
Family history of AMI	5 (2.4%)	12 (4.0%)	9 (3.4%)	0.643
Arrhythmia, n (%)	20 (5.4%)	32 (6.7%)	24 (6.2%)	0.710
Smoke, n (%)	66 (32.2%)	118 (39.1%)	76 (29.1%)	0.038
Drinking, n (%)	26 (12.7%)	43 (14.2%)	20 (22.5%)	0.044
Avg SBp, mmHg	115.8 ± 15.0	113.6 ± 15.3	112.5 ± 15.9	0.398
Avg DBp, mmHg	66.6 ± 8.1	66.1 ± 9.0	66.1 ± 8.9	0.568
AKI, n (%)	20 (4.9%)	78 (18.9%)	134 (32.5%)	<0.001
KDIGO stage, n (%)				<0.001
1	19 (4.6%)	62 (15.0%)	79 (19.2%)	
2–3	1 (0.3%)	16 (3.4%)	55 (14.2%)	

Abbreviation: Hx: history, HTN: hypertension, Avg SBp: average systolic blood pressure, Avg DBp: average diastolic blood pressure.

**Table 3 biomedicines-13-01909-t003:** Comparison of clinical characteristics and laboratory parameters of AMI patients.

Variables	T1 (FAR < 3.48) n = 412	T2 (3.48 ≤ FAR ≤ 5.10) n = 412	T3 (FAR > 5.10) n = 412	*p*
BMI, kg/m^2^	23.80 ± 3.16	23.92 ± 3.74	23.14 ± 3.79	0.139
T3, nmol/L	1.06 ± 0.61	1.01 ± 0.28	0.95 ± 0.34	0.018
T4, nmol/L	7.48 ± 2.00	7.35 ± 2.09	7.20 ± 2.62	0.328
Hcy, mmol/L	18.00 (14.10, 27.20)	18.10 (14.30, 26.40)	19.60 (15.63, 27.03)	<0.001
LDL-C, mmol/L	2.35 ± 0.89	2.38 ± 0.88	2.31 ± 0.88	0.585
CHOE, mmol/L	4.09 ± 1.13	4.08 ± 1.06	3.92 ± 1.08	0.070
HbA1c, %	6.23 ± 1.34	6.52 ± 1.53	6.75 ± 1.70	<0.001
cTnT, ng/mL	0.42 (0.08, 1.31)	0.54 (0.11, 1.66)	0.64 (0.12, 2.07)	0.043
NT-proBNP, pg/mL	659.25 (249.48, 1821.25)	844.30 (248.35, 2300.25)	1860.00 (563.53, 5356.25)	<0.001
LAC, mmol/L	2.311 ± 1.17	2.59 ± 1.95	2.54 ± 1.95	0.177
AB, mmol/L	23.54 ± 1.93	23.36 ± 2.56	22.66 ± 3.29	0.001
pH, mmol/L	7.41 ± 0.05	7.41 ± 0.05	7.39 ± 0.06	0.003
CK-MB, U/L	35.2 (16.2, 100.6)	29.8 (15.0, 88.7)	29.7 (14.4, 74.4)	0.639
CK, U/L	304.0 (118.0, 1012.0)	220.0 (106.0, 786.0)	250.5 (107.5, 735.3)	0.519
P^−^, mmol/L	0.87 ± 0.38	0.95 ± 0.36	1.07 ± 0.45	<0.001
Mg^2+^, mmol/L	0.97 ± 0.13	0.99 ± 0.14	1.01 ± 0.16	0.001
Ca^2+^, mmol/L	2.21 ± 0.19	2.24 ± 0.16	2.19 ± 0.18	0.001
Cys-C, mg/L	0.94 ± 0.34	1.04 ± 0.39	1.40 ± 0.88	<0.001
GLU, mmol/L	7.73 ± 4.02	8.56 ± 4.20	9.44 ± 5.26	<0.001
D-D, mg/L	1.24 ± 5.28	1.42 ± 4.59	2.51 ± 6.76	0.006
hs-CRP, mg/L	3.37 ± 3.67	3.72 ± 2.69	8.46 ± 4.51	0.410
Neu, 109/L	7.11 ± 3.59	7.43 ± 3.43	8.17 ± 4.26	0.000
Lym, 109/L	1.48 ± 0.86	1.54 ± 0.91	1.44 ± 0.76	0.198
HGB, g/L	138.30 ± 19.37	138.06 ± 19.13	130.22 ± 22.13	<0.001
RDW, %	13.04 ± 1.07	13.18 ± 1.10	13.36 ± 1.11	<0.001
AST, U/L	50.0 (27.0, 128.0)	46.0 (27.0, 103.0)	48.5 (26.3, 102.5)	0.155
ALT, U/L	34.0 (23.0, 53.2)	33.0 (22.0, 50.0)	35.0 (22.3, 57.5)	0.004
LVEF, %	51.6 ± 10.9	51.6 ± 11.5	48.6 ± 12.5	0.003
LVSD, mm	50.78 ± 8.44	51.31 ± 7.46	52.37 ± 7.21	0.018
LVDD, mm	37.33 ± 6.84	37.33 ± 7.46	39.04 ± 8.03	0.087

Abbreviation: FAR: fibrinogen-to-albumin ratio, Hcy: homocysteine, CK-MB: creatine kinase-MB isoenzyme, CK: creatine kinase, cTnT: cardiac troponin T, NT-proBNP: N-terminal pro-B-type natriuretic peptide, hs-CRP: high-sensitivity C-reactive protein, LDL-C: low-density lipoprotein cholesterol, HbA1c: hemoglobin A1c (glycated hemoglobin), GLU: Glucose, CO_2_: carbon dioxide, Ca^2+^: calcium, K^+^: potassium, Mg^2+^: magnesium, P^−^: inorganic phosphate, AB: actual bicarbonate, Cys-C: cystatin C, BMI: body mass index, CHOE: total cholesterol, HbA1c: hemoglobin A1c, T3: triiodothyronine, T4: thyroxine, D-D: D-Dimer, Hcy: homocysteine, LAC: lactate, Neu: neutrophil count, Lym: lymphocyte count, HGB: hemoglobin, RDW: red blood cell distribution width, AST: aspartate aminotransferase, ALT: alanine aminotransferase, LVEF: left ventricular ejection fraction, LVSD: left ventricular systolic dimension, LVDD: left ventricular diastolic dimension.

**Table 4 biomedicines-13-01909-t004:** Multivariate regression of acute kidney injury.

	Model 1		Model 2		Model 3	
	OR (95%CI)	*p*	OR (95%CI)	*p*	OR (95%CI)	*p*
FAR	1.465 (1.362, 1.577)	0.000	1.415 (1.313, 1.525)	0.000	1.539 (1.352, 1.750)	0.000
Age, year	0.990 (0.940, 1.042)	0.701	-		-	
Male, n (%)	0.881 (0.589, 1.317)	0.000	-		-	
HX of Hypertension	0.651 (0.440, 0.965)	0.032	0.630 (0.425, 0.933)	0.021	-	
HX of T2DM	1.210 (0.777, 1.882)	0.399	-		-	
K^+^	-		1.657 (1.161, 2.366)	0.005		
RDW	-		1.214 (1.049, 1.404)	0.009		
P^−^	-		-		1.262 (0.521, 3.056)	0.606
apoA	-		-		6.869 (1.351, 34.927)	0.020
LVEF	-		-		0.954 (0.927, 0.981)	0.001
T3	-		-		0.589 (0.202, 1.718)	0.332

Model 1: adjusted for age, sex, history of hypertension, history of T2DM; Model 2: adjusted for history of hypertension, K^+^, RDW; Model 3: adjusted for apoA, P^−^, T3, LVEF. K^+^: potassium, RDW: red cell distribution width, apoA: apolipoprotein A, LVEF: left ventricular ejection fraction, T3: thyroxine.

## Data Availability

The datasets for this study are available on request to the corresponding author.

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
