# Peer review of "Fibrinogen-to-Albumin Ratio Predicts Acute Kidney Injury in Very Elderly Acute Myocardial Infarction Patients"

_biomedicines, 2025, doi:10.3390/biomedicines13081909_

Round 1
Reviewer 1 Report
Comments and Suggestions for Authors
The manuscript addresses an important clinical issue: the prediction of acute kidney injury (AKI) in very elderly patients with acute myocardial infarction (AMI). The use of the fibrinogen-to-albumin ratio (FAR) as a potential biomarker is both clinically relevant and methodologically feasible.
The authors present a well-structured retrospective study with solid statistical analyses, however several aspect have been improved. Following major comments:
- Could you further clarify why patients ≥75 were chosen specifically, rather than a broader elderly cohort (e.g., ≥65), and how this cutoff impacts clinical applicability?
- Could you justify more clearly the choice of a retrospective single-center study design in light of the generalizability of the findings? How did you address potential biases in data collection and AKI diagnosis timing?
- What proportion of initially screened patients were excluded due to incomplete data or comorbid conditions, and could this have introduced selection bias?
- Was the timing of fibrinogen and albumin measurement standardized relative to AMI onset or admission? How might variability in timing affect FAR’s predictive accuracy?
- What could explain the seemingly paradoxical finding that a history of hypertension was inversely associated with AKI?
- Were sensitivity, specificity, and predictive values calculated for the optimal FAR cutoff, and could these be reported explicitly?
- In addition to your study’s clinical implications, you might consider linking your findings with recent developments in biomedical AI. For instance, works such as 10.1007/978-3-031-57430-6_20 demonstrate how recurrent neural networks and CNNs have been applied effectively to cardiovascular signals and clinical imaging diagnostics. Could future iterations of your model incorporate such methods to enhance FAR-based prediction?
The English could be improved to more clearly express the research.
Reviewer 2 Report
Comments and Suggestions for Authors
This study evaluates the predictive utility of the fibrinogen-to-albumin ratio (FAR) for acute kidney injury (AKI) in elderly patients with acute myocardial infarction (AMI). AKI is a frequent complication of AMI—particularly in older individuals—and reliable early-warning markers are lacking. In a retrospective cohort encompassing AMI admissions to the First Affiliated Hospital of Xi’an Jiaotong University from 2018 to 2022, the investigators examined the relationship between FAR and subsequent AKI. Elevated FAR was independently associated with a markedly higher risk of AKI, demonstrating robust discriminative performance, with especially strong predictive power for stage 2 AKI. Owing to its objectivity and routine availability, FAR may constitute an effective early biomarker for AKI surveillance in elderly AMI patients. The authors advocate prospective studies to validate these findings and to investigate potential therapeutic interventions informed by FAR-guided risk stratification. However, a couple of issues should be addressed and solved.
- The multivariate regression revealed that a prior history of hypertension was inversely associated with acute kidney injury (AKI), indicating a “protective effect” (odds ratios = 0.651 and 0.630 in Models 1 and 2, respectively). The Discussion explicitly states that “the paradoxical protective effect of hypertension warrants further mechanistic investigation”. To deepen the manuscript, it would be valuable to provide a more detailed exploration of potential explanations for this paradoxical observation.
- The results show that FAR achieved an AUC of 0.79 for predicting overall AKI, whereas its discriminatory ability rose markedly to an AUC of 0.94 (95 % CI, 0.92–0.97) for predicting stage 2 AKI². This is a highly significant finding, yet the Discussion could delve more deeply into the clinical implications of this disparity.
- The authors shold discuss the study’s limitations and outline potential avenues for improvement.
Round 2
Reviewer 1 Report
Comments and Suggestions for Authors
The authors have been addressed all my comments.
Reviewer 2 Report
Comments and Suggestions for Authors
The revised manuscript has been largely improved and addressed my concerns. I have no problem to endorse its publication.